# Green accounting and reporting in Bangladesh's pharmaceutical and textile industries: A holistic perspective

**Mohammad Mizenur Rahaman**[1], **Shamima Akter**[1], **Md. Alamgir Hossain**[2], **Adiba Rahman Bushra Chowdhury**[1], **Renhong Wu**[3]*

**1** Department of Business Administration, Shahjalal University of Science and Technology, Sylhet, Bangladesh, **2** Department of Management, Hajee Mohammad Danesh Science and Technology University, Dinajpur, Bangladesh, **3** School of Management, Kyung Hee University, Seoul, Republic of Korea

* wurenhongbini@163.com

**Data Availability Statement:** All data are in the supporting information files.

**Funding:** The author(s) received no specific funding for this work.

## Abstract

This study examined the factors influencing green accounting and reporting practices (GARPs) in Bangladesh's pharmaceutical and textile industries. Hence, it draws upon disclosure theory to disclose relevant information in the context of environmental accounting and encourages them to boost their environmental performance. It utilized content analysis from 13 pharmaceuticals and 22 textiles data from Dhaka stock exchange (DSE) listed companies of Bangladesh and applied quantitative methods for comparative analysis. The findings showed that GARPs are influenced by firm characteristics and external factors rather than organizational performance, and eleven environmental indicators (separately) have a lower mean of less than 0.50 in both industries. Firms' general characteristics (FFGC) are noteworthy factors that exhibit a negative coefficient for both the pharmaceutical and textile sectors but hold a robust impact on the GARPs, with P = 0.007 and 0.003, respectively. The statistical significance of environmental factors (EFs) applies to the textile sector p = 0.000. Implementing GARPs in the pharmaceutical industry proved more effective than in the textile sector, offering valuable support to managers in expediting environmental practices in Bangladesh's textile industry.

## 1. Introduction

The topic of garments and pharmaceuticals as emerging contaminants in aquatic environments is widely debated on a global scale. Bangladesh, like many other countries, faces a significant threat from pharmaceutical pollution, particularly in its rivers. The impact of climate change further exacerbates existing challenges by disrupting food production systems and reducing fish productivity under harsh environmental conditions. In Malaysia, climate change directly affects agriculture in several ways. Firstly, it leads to a decline in agricultural productivity [1]. Additionally, it contributes to increased food insecurity and protein deficiency due to elevated prices. The high temperatures of both air and water result in economic costs by reducing system efficiency and power generation [2]. Furthermore, changes in precipitation,

**Competing interests:** The authors have declared that no competing interests exist.

temperature, and cloud cover, and increased carbon dioxide concentrations alter the geographic distribution and crops, imposing modified boundaries [3]. Textile industries contribute to 20% of the global water pollution, resulting in detrimental effects on ecosystems through water surface acidification and eutrophication [4]. Another example of how the trade-off between sustainability, efficiency, and effectiveness goals is a clear manifestation of the management complexity that impacts pharmaceutical firms and the health systems overall [5]. The requirement for the pharmaceutical sector to supply a greater choice of pharmaceutical products while pursuing sustainable development is being driven by two factors: an aging global population and enhanced healthcare systems [6]. The research will close this gap by examining how green accounting techniques, especially in the environmental and economic spheres, might enhance environmental performance and energy efficiency. According to [7,8], businesses should prioritize green accounting practices and coordinate their economic, environmental, and social policies with the Sustainable Development Goals (SDGs) of the United Nations and other environmental criteria. There is no way to reduce the production level, and the growing industries without proper sustainability would be a disaster with current trends in mind, this study projects GHG emissions through 2030 (26 thousand Gg CO2 eq.). By 2050, the world's consumption of textiles and clothing is predicted to triple due to present patterns in consumer demand [9]. If the sustainability did not ensured the performance will not be measured in actual term. This study will fill the gap of theoretical literature and empirical evidence on supporting of EARP to disclose the materiality, as stakeholder institutions find it challenging to measure the true effects because of complicated downstream supply chains and a dearth of reliable data in manufacturing industry like Pharma and garments.

During the period of 2020–21, the total fish production reached 46.21 lac Metric Ton (MT). Notably, the production from closed water bodies has witnessed a significant increase due to the dissemination of adaptive technologies and the provision of need-based extension services. However, there has been a slight decline of 0.5% in the productivity of Hilsha fish from 2020 to 2021[10]. Moreover, in the city of Barishal, researchers have discovered alarming levels of the antibiotic metronidazole, exceeding the safe target by more than 300 times [11]. This poses a serious concern for the environment and public health in the region. Additionally, a concerning number of 149 rivers located near industrial areas, have been found to be heavily polluted with garment wastewater and out of the 2,220 factories located in the industrial district of Gazipur, only 556 have implemented Effluent Treatment Plants (ETPs), indicating that the existing (ETPs) are failing to adequately safeguard the environment [12]. The discharge of pharmaceutical compounds into rivers has led to significant adverse impacts and, in some cases, even posed a threat to the survival of native fish species due to excessive exposure. Shockingly, it is concerning that less than 1% of factories in Bangladesh utilize or maintain Effluent Treatment Plants (ETPs). This lack of implementation is compounded by the limited capacity of experts in river basin management, sustainability, and delta studies to effectively monitor ETPs within factories.

Green accounting or sustainable accounting, have become increasingly challenging to make a meaningful contribution to environmental factors and firm performance, which is essential for articulating organizational strategies and making investment decisions [10]. By 2030, concerted efforts are required to eliminate poverty, safeguard the environment, and foster universal peace and prosperity, but the funding for climate action is only 8.1% [13]. United Nations (UN) sustainable development goals serve as guidelines and targets that every nation must follow to fulfill its developmental ambitions [14]. But, industries internal control and disclosure are highly susceptive to increase the water level pollution and negative impact on aquatic ecosystem. As a result, Environmental Accounting (EA), including the disclosure of environmental aspects in annual reports and other company publications, has emerged as a

crucial component of modern corporate reporting [15]. The government of Bangladesh considers the pharmaceutical industry a crucial sector for generating revenue, and it plays a significant role in contributing 1.0% to the country's gross domestic product. In the second quarter of 2022, the industry's value reached TK 265.95 billion [16]. The overall sector continues to face challenges with backward linkage, which remains a weak point. It presents a potential threat to the ecological system due to their extensive utilization of natural resources, resulting in the emission of greenhouse gases and wastewater. Similarly, the textile industry is renowned for its significant contribution to water pollution, primarily attributed to its extensive water consumption and discharge of coloring materials in the effluent and the catastrophic impact of textile dyes on human health and the environment [17].

Prior studies affirm that creating a new index based on a secondary dataset would be more beneficial for analyzing sustainable development policies [18]. Again, several studies investigated the disclosure of Environmental Accounting Reporting Practices (EARPs); this research makes a ground for presenting the rank and environmental aspect of Bangladeshi leading manufacturing industries. However, Bangladesh still needs to be more incognizant in this research issue. Hence, there is a scope to address the gap of more environmentally sensitive industries in Bangladesh's pharmaceuticals and textile industries through a comparative analysis unseen in the previous literature. This study followed stakeholder and disclosure theory and can fill this gap by identifying the factors affecting the EARPs and determining the level of EARPs in the manufacturing sector, especially in the pharmaceutical and textile industries in Bangladesh, which will support the policymakers and company executives to make close eyes on these factors to attract the foreign investors.

## 2. Literature review and hypothesis development

To survive and expand in today's economy, businesses increasingly focus on streamlining their operations and maximizing the resources at their disposal. To ensure this, they specialize in environmental aspects to scale back environmental impacts and improve environmental performance. In recent days, the corporate sector of Bangladesh has been concerned about EA practices, but they have not been performing as well. Numerous papers have already explored the use of developing economies as samples, such as Gola [19], Leung [20], Dhar and Chowdhury [21], Masud et al. [22], Ribeiro et al. [23], Kabir and Akinnusi [24], and Jones [25], among others. Similarly, many studies have been conducted on environmental disclosure and EA practices in developed countries [23,26,27]. The organization's environmental performance draws particular attention to internal and external stakeholders, and a big challenge and concern is delivered to the accounting and sustainability of the international organization [28]. The maintenance of firms is increasingly hooked on social responsibility, which has caused businesses to report on their sustainability and EA [29].

On the one hand, environmental protection prevents environmental disasters and safeguards the environment, while on the other hand, it fosters economic development. However, Bangladesh's textile industry's green accounting reporting practice could be better and more satisfactory [30]. Only a few Bangladesh corporations are voluntarily attempting to offer poor-quality social and environmental data [31].

### 2.1 Reviewing the prior literature on Bangladesh

The environment has suffered from careless industrial practices, consumer apathy to the environmental repercussions of rising consumerism, and economic inequality across societal segments in Bangladesh. A lot of emphasis has been paid to the economic aspect of sustainability, particularly in the past 20 years. This indicates a research void that the paper aims to bridge.

Sustainable accounting, energy consumption, and environmental sustainability, especially as they relate to chemical and pharmaceutical businesses operating in developing nations such as Bangladesh [8]. Pharmaceutical and chemical industries in Bangladesh use particular green accounting techniques, such as emission tracking, which involves the measurement and monitoring of greenhouse gas (GHG) emissions and other airborne pollutants by the companies, [32]. Another approach is waste management and minimization, in which businesses evaluate the production and disposal of hazardous waste and put strategies into place to lower trash generation, enhance disposal techniques, and recycle waste when it is practical [33]. It has been discovered that Green Human Resource Management has a positive impact on job performance and job satisfaction because it increases employee engagement and motivation to enhance organizational performance in Bangladeshi pharmaceutical companies. Additionally, GHRM advances the theories of action, job performance, and ability to inspire [34]. Results showed that business efficiency and environmental compliance have a significant and positive impact on sustainability. They also showed that environmental costs mediated the effect of ecological accounting on sustainability in pharma enterprises in Ghana [35]. Similarly, due to institutional pressure, waste management had a positive and significant impact on the environmental sustainability of small and medium-sized businesses [36]. Energy-efficient practices can be encouraged as a practical way for manufacturing enterprises to achieve sustainability, and energy-efficient practices can improve green tax policies [37]. Vertically, customers that care about the environment are pushing the textile sector to reveal the environmental impact of their products throughout their life cycles and are placing greater demands on sustainable products [38–40]. This study offers insightful information about the complex relationships that exist between environmental accounting, financially successful production, and sustainable practices in the pharma textile sector and favorably affect financial performance without having a greater impact on sustainable production in textile sector.

## 2.2 Theoretical literature

Many theories have been examined in EARP, Institutional theory defines that an organization's behavior and practices can be influenced by the external environment, which includes stakeholder norms, values, and expectations [41]. The application of institutional theory can offer a valuable perspective in comprehending the impact of institutional forces on the adoption of green accounting practices and energy efficiency measures in Bangladesh, particularly in the chemical and pharmaceutical sectors [8].

In the disclosure literature, theories have been established to account for the rationale behind the choice to provide further information. The results obtained are inconsistent, and the empirical evidence does not consistently support disclosure theories [42]. It has two main objective, firstly, to validate the factors used and how the factors influences results. Whereas, the voluntary disclosure theory's (VDT) assumptions, financial control variables are being included in an increasing number of environmental disclosure studies [43]. Alternatively, Sheth [44] have distinguished between two distinct facets of the economic dimension of sustainability, namely external stakeholders and cost reduction. Several studies resulted, favorable correlation between CSR disclosure and environmental performance, bolstering the voluntary disclosure idea [45,46]. Another fact is "transparency fallacy", when corporate transparency is forced upon it by external stakeholders; transparency should instead be linked to increased engagement with environmental issues and ethical business practices [47]. In order to strategically reduce information asymmetries in their business behavior and dispel concerns about their societal legitimacy, firms use sustainability reporting as a crucial tool of corporate communication. It suggest that companies that perform better in terms of the environment have

an incentive to release more information to external stakeholders. This is done to communicate the firm's other positive attributes and unobservable proactive strategy. In this study disclosure theory used, to identify the GRI parameters are fully supported and the EARP have positive impact on financial performance.

Stakeholder pressure is a crucial component of green accounting. To succeed and maintain ecological sustainability, relationships between businesses and stakeholders are essential. According to [48], stakeholder theory implies that companies should take into account the expectations and interests of all parties involved, not just shareholders. Employees, clients, suppliers, investors, regulators, and the larger society in which the business operates are examples of stakeholders [49]. According to the theory, companies can generate value by attaining sustainable long-term success while striking a balance between the demands and expectations of many stakeholders [50]. The stakeholders of pharmaceutical and chemical industries in Bangladesh are diverse and include workers, clients, financiers, vendors, and legislators [51]. Companies must strike a balance between these stakeholders' divergent expectations and interests if they are to flourish. Employees could be worried about having a safe and healthy workplace, for instance. Ecological disclosure is typically inadequate whenever participants are not involved [52]. Previous research has demonstrated that information discrepancies can be reduced when a business shares financial information and other details with its stakeholders [53]. Furthermore, one of the main forces behind how businesses respond to ecological concerns is stakeholder pressure for greener operations and strategies in the companies where they operate [54]. There is a dearth of research on the impact of stakeholder pressure on ecological sustainability, with the majority of studies being carried out in industrialized countries. Pressure from stakeholders has a significant impact on implementing green management and sustainability performance, as Khare et al. [55] found. Similarly, Latifah and Soewarno [36] discovered a favorable correlation between sustainability and normative pressure from interested parties. As per [56], firms are no longer just driven by economic expansion as their motivation. Important roles are also played by elements of the natural world and civilization. Only with the active engagement and cooperation of all pertinent parties can sustainable development take place [35].

## 2.3 Environmental costs

Environmental accounting's primary purpose is to make managers aware of the monetary costs associated with environmental damage, which encourages them to find ways to lessen those costs and their company's overall influence on the environment. Managers must account for environmental costs in the organization. As a result, inspiring them to decrease and avoid environmental costs while lowering the company's environmental effect is a win-win situation [57,58]. Organizations use Environmental Management Accounting (EMA) to identify ecological expenses [59], and managers and accountants focus on managing environmental costs to monitor the firm's environmental performance.

## 2.4 Environmental discloser

Corporate social and environmental disclosure may not apply to all countries in the same way. Environmental disclosure, however, varied depending on whether a company's environmental operations were classified as good, mixed, or inadequate [60]. Some researchers suggested a requirement to review the explanations behind organizations adopting environment-related management accounting procedures as they might be applied at a firm, regional, or national level [61–64]. Other financial characteristics, such as decentralization [65]; legal system [66];

stock market listing, industrial type [67]; high polluting industries, and environmental performance [15], have a favorable link with the level of disclosure of environmental information.

## 2.5 Global Reporting Initiative (GRI) guidelines related to environmental aspects

Researchers can use GRI guidelines to examine organizations' disclosure procedures and conduct environmental practices in an internationally standard format. EARPs Indicators and these principles are suggested to be practiced by all manufacturing or other corporate organizations mandatorily to better the environment. These standards cover a total of twelve environmental-related topics. Material, Energy, Water, Transportation, Biodiversity, Emissions, Effluent and Waste, Products and Services, Compliance, Overall Environmental Costs, Supplier Environmental Assessment, Environmental Grievance Mechanisms, and Occupational Health and Safety are just a few of the categories (GRI, 2018) and connection with the disclosure of both environmental and social approaches are analyzed by [68].

## 2.6 EARPs

In both developed and developing countries, social and environmental reporting has increased during the last two decades [24,31,69]. Ideas have been proposed to explain the incentives for implementing EARPs from a theoretical approach. Even though the majority of empirical studies on EA have focused on private companies [60,70–73], there has been a gradual trend in recent years towards the study of public entities' environmental accounting practices, such as Australia [74–76]. Overall, the findings of these researches show that environmental reporting in government agencies has risen over time. According to previous research, internal factors and maintaining hygine at the institutional level are significant elements of environmental accounting reporting procedures [77], and resulting green accounting has a noteworthy and favorable influence on both energy efficiency and environmental performance. But, in developing nations, the impact is not the same, i.e, behavior of the environmental Kuznets curve (EKC), which is probably empirically the most established regularity within the framework of environmental macroeconomics [78,79]. Additionally, the correlation between green accounting and environmental performance is partially mediated by energy efficiency [80].

## 2.7 Firms Features (FFGC) and EARPs

In the pharma industry, competition and free cash flow capacity play a vital role in maintaining Environmental Practice (EP) [10]; its financial leverage hampers the activity of its process [81]. The association between environmental information given and corporate characteristics was explained by several authors. A list of factors, i.e., Size, category, culture, CSR, and Porition, has been studied for a long time. But Strategic attitude, Managers' perception, risk management, and corporate governance now contribute to organization performance attributes. Several studies have discovered that firm size has a large and favorable impact on EARPs and the level of voluntary information disclosure [15,65,67,82]. Christ and Burritt [83], and Ofoegbu and Megbuluba [84] focused on company size as a factor in how easily Environmental Accounting (EA) can be applied in businesses; specifically, the higher the enterprise's income, the easier it will be to apply EA. Implementing a code of conduct by corporations may be regarded as a strategy for legitimization [85–87].

*H1*: *There is a relationship between the General Characteristics of the firm and Environmental Accounting and Reporting Practices.*

## 2.8 Firms performance and EARPs

Firm performance is one of the important components of EARPs and is defined as profitability and environmental transparency, which have yielded a mixed bag of results [65,88,89]. Moreover, improving profitability and reducing financial leverage are two strategies for encouraging the disclosure of environmental accounting data [27] and environmental cost management accounting performance and business performance. Nonetheless, other studies have concluded that a company's financial performance has little bearing on the amount of social and environmental disclosures made [85,90–92].

*H2: There is a relationship between Firm Performance and Environmental Accounting and Reporting practices.*

## 2.9 Stakeholder Pressure (SP) and EARPs

Community stakeholders have a major bearing on how and what businesses produce and do in their local communities [93]. Jamil et al. [94] highlight the importance of association or media; government agency demands on firms to use Environmental disclosure is required since it may aid decision-making and control by stakeholders, as well as facilitate incentive mechanisms [27]. EA to meet environmental protection criteria, which is one of the variables influencing the adoption of EA in organizations. Bangladesh needs to pay more attention in EARP investigations compared to other developing countries. There needs to be more organizational commitment in Bangladesh to the various stakeholders involved in societal activities, and Stakeholder pressure in the form of investor pressure, media pressure, and staff pressure is justified [58].

*H3: There is a relationship between Stakeholder Pressure and Environmental Accounting and Reporting Practices.*

## 2.10 External factors (EF) and EARPs

The management team participates in performance evaluation, visualization, intellectual capital, and employee empowerment. It has been observed that the correlation between hard lean practices and performance is positively moderated. Conversely, when managers prioritize goal setting and work standardization, performance outcomes are diminished [95,96]. For instance, changes in environmental regulations could have a significant influence on the entity's location [75], funding source [76], management attitudes toward environmental protection [97], and community expectations regarding the local entity's environmental performance [98], influence the extent to which entities develop EARPs [99]. Increasing recognition is given to the role of institutional elements of the environment in determining the structure and performance of organizations [100]. More financial resources are needed for SMEs to establish environmental management systems in their operations [101] despite rising environmental awareness and disclosure [94]. Governments, other regulatory agencies, and businesses are concerned about EA [102] and provide environmental data in their yearly reports. The effectiveness of EMA is greatly affected by coercive forces [94]. Therefore, businesses are less likely to adopt EMA on their own will if the government does not mandate its use [103]. The national economy of Bangladesh tends to overlook the significance of green foods and services. In numerous instances, the role of the environment in enhancing the country's economic sustainability is disregarded. However, the green reporting practice in Bangladesh's textile industry is relatively low and unsatisfactory [30].

*H4*: *There is a relationship between External Factors and Environmental Accounting and Reporting Practices.*

## 2.11 Conceptual framework

After looking at the literature, this study develops a conceptual framework (Fig 1), showing the relationship among the variables (Independent and dependent variables).

## 3. Methodology

The population of this study comprised 32 pharmaceutical firms and 55 textile firms listed on the Dhaka Stock Exchange (DSE). However, the inclusion of the companies was conditional on the company being listed on the DSE for at least five consecutive years between 2017 and 2021. In determining the final sample, one extra criterion was used: the firms had to have at least five years of annual reports to allow for a longitudinal study. 13 pharmaceuticals and 22 textiles firms were selected considering the performance of last year within the industry, accounting for 40% of the total DSE listed firm. Moreover, a minimum acceptable sample size was considered to be 10% of the total population [51]. Thus, 35 firms listed on the DSE were used for the study. Hence, the total sample size is 175 annual reports for the 35 firms. Data before 2017 was not considered as EARPs was not mentionable practice in Bangladesh.

### 3.1 Econometric approach

To determine the factors of EA and its effect on EARPs, this study uses the statement of EARPs using the Multiple Linear Regression Model. The Multiple Linear Regression Model has been postulated as follows:

$$Y = \beta_\circ + \beta_1 X_1 + \beta_2 X_2 + \beta_3 X_3 + \beta_4 X_4 + \beta_5 X_5 + \beta_6 X_6 + \varepsilon$$

$$\text{So, } EARPs = \beta_\circ + \beta_1 FFGC + \beta_2 ROA + \beta_3 ROE + \beta_4 ROI + \beta_5 SP + \beta_6 EP + \varepsilon$$

Where,

*EARPs* for Environmental Accounting Reporting Practices; *FFGC* for Firm general characteristics; *ROA*, *ROE*; *ROI*; SP for Stakeholder Pressure, and EP for External pressure is the intercept and $\beta_1, \beta_2, \beta_3, \beta_4, \beta_5, \beta_6$ are regression co-efficient. Here, the intercept term or regression constant $\beta_\circ$ means the mean value of the dependent variable Y when independent variables have no impact.

### 3.2 Data collection procedure and variable description

This study involved using annual reports as the primary source of information concerning environmental sustainability issues. Annual reports were chosen due to their broad credibility, widespread acceptance, and large audience, as highlighted in previous works [104,105]. A content analysis of the annual reports was conducted to investigate the environmental accounting practices of the companies. Content analysis is widely recognized as the most suitable method for scrutinizing corporate reporting practices [106,107].

The study created an evaluation checklist of tools to assess the level of green accounting practices, based on the Global Reporting Initiative (GRI) IV categories. Qualitative content analysis was employed to collect data from selected annual reports, adhering to GRI guidelines, which served as a benchmark for measurement [108,109]. A similar study by Hossain et al. [68] utilized a mixed methods approach, incorporating descriptive statistics, correlation, and multiple regression. Following the analysis of environmental information in the annual

**Firm Features:**
**A: General Characteristics**
Firms Size
Category Status
Organizational Culture
Corporate Social
Responsibilities
Position in the Value Chain
Strategic Attitudes
Motivation of Managers'
Attitudes
Risk Management
Corporate Governance
**B: Financial Resources**
Total Assets
Total Investments
Total Debt
**C: Firms Performances**
Return on Assets
Return on Equity
Return on Investments
Earnings per Share

**Stakeholder Pressure:**
Organizational Regulation
Media
Pressure from Employees
Pressure from Investors

**External Factors:**
Social Environmental
Impact
Production Process
Firms Location
Global Culture
Regulatory Environment

**Environmental Accounting**
**& Reporting Practices:**

- Energy
- Effluents & wastes
- Products & Services
- Materials
- Suppliers Environmental
  Assessment
- Biodiversity
- Emission
- Water
- Environmental
  Compliances
- Environmental Grievance
  Mechanism
- Occupational Health &
  Safety

**Fig 1. Conceptual model on effects of environmental factors on EARP.**

reports, an interpretative and critical textual analysis of the reports was undertaken to identify specific themes. These themes emerged through interpretative analyses involving observation, reading, and reviewing the reports. 26 environmental sustainability data points, as outlined in the Appendix (Tables A3 and A4 in S2 File), were identified. Furthermore, the environmental sustainability information scores for each company were expressed as mean scores.

Consistent with studies by Maama and Appiah [104], Bravo and Reguera-Alvarado [110], and Van Zyl [111] the recording of green accounting practice was conducted using a dummy variable, referred to as green accounting practice (GAP). A value of 1 represented a firm practicing GAP, while 0 denoted otherwise, based on the perceived usefulness of the disclosed information. The assessment of disclosure usefulness was grounded in the extent to which the information aligned with the qualitative characteristics of accounting information, as specified in the conceptual framework of the IFRS and the guiding principles of GRI IV.

The quality and quantity of environmental information presented in annual reports were used to assess the firms' environmental accounting, resulting in an environmental accounting disclosure quality score. These scores for each information category were added together to produce an overall environmental accounting score that was consistent with earlier research [110,111].

### 3.3 Validity and reliability of the green accounting factor scores

The authors coded annual reports, encompassing a total of 175 reports. To ensure consistency and reliability in the coding process, the authors thoroughly examined and discussed pertinent literature at the outset. Subsequently, detailed guidelines were formulated to facilitate the reading of annual reports and the corresponding coding of the evaluation matrix. Following coding all 175 annual reports, the Kappa intercoder reliability rate was computed.

Typically, the Kappa value ranges between 1 (indicating perfect agreement) and 0 (indicating no agreement). According to Stevens et al. [112], a Kappa value of 0.5 signifies moderate agreement, 0.7 denotes good agreement, 0.8 indicates very good agreement, and 0.85 represents excellent agreement. The computed average intercoder reliability was 2.56%, a statistically significant result at the 0.01 level. Notably, these findings surpassed the 81% benchmark widely endorsed by researchers, as stipulated by Stevens et al. [112].

## 4. Results and analysis

### 4.1 Descriptive statistics

The present study measures the descriptive statistics of dependent and independent variables of companies' pharmaceutical and textile industries presented in the Appendix (Tables A1 and A2 in S2 File). Appendix A (Tables A1 and A2 in S2 File) explains the pharmaceutical and textile industry's descriptive statistics. This study found that all the pharmaceutical and textile industry indicators of the dependent variable of environmental aspects (Energy, Effluents and Waste, Products and Services, Materials, Supplier Environmental Assessment, Biodiversity, Environmental Compliance, Health and Safety, Water, Grievance Mechanism and EARP) showed a lower mean of less than 0.50.

Furthermore, to show all the environmental aspects or practices in one variable, this study has accumulated the eleven variables to the EARPs. The mean value of this variable is 0.30, and the standard deviation is 0.197 in the textile industry. Following, the mean value of EARPs is 0.36, and the standard deviation, is 0.196 in the pharmaceutical industry. The range of the maximum and minimum values of both the textile and pharmaceutical industries Cumulative EARP are subsequently sated between 0.81 and 0.06 and 0.00 and 0.81, respectively.

Consequently, all the independent variables of the study showed mixed results. As per the analysis and by reviewing the annual reports, it is found that in the textile and pharmaceutical industry, the mean value of the FFGC is 0.66 and 0.82, and the standard deviation is 0.305 and 0.284. Therefore, all the indicators of firm performance variables, i.e., ROA, ROE, ROI, also showed mixed results. Finally, the variables have mean values for both textile and pharmaceuticals SP 0.38 and 0.52, EF 0.86, respectively. Similarly, in the textile industry, all the study's

Table 1. Correlation analysis of pharmaceutical companies.

| Variables Name | FFGC | ROA | ROE | ROI | SP | EF | EARPs |
|---|---|---|---|---|---|---|---|
| FFGC | 1 | | | | | | |
| ROA | 0.245* | 1 | 1 | 1 | 0.351** | 1 | 1 |
| ROE | 0.225 | 0.919** | 0.932** | 0.353** | 0.560** | 0.466** | |
| ROI | 0.245 | 0.984** | 0.497** | 0.242 | | | |
| SP | 0.420** | 0.334** | 0.233 | 0.128 | | | |
| EF | 0.709** | 0.243 | 0.169 | | | | |
| EARPs | 0.602** | 0.159 | | | | | |

independent variables reveal mixed results. However, the comparative analysis of descriptive statistics shows that textile companies have poorer results than pharmaceutical companies in every aspect. Similar result found in that EP have significant impact ($R^2 = 0.61$) on green accounting practices have a significant positive effect through economic and environmental practices and EF partially mediates the relationship between green accounting and environmental performance [8].

## 4.2 Correlation analysis

The correlation matrix shown in Tables 1 and 2 emulates the pair-wise correlations between all the combinations of two of the variables in this study. Pearson product-moment correlation coefficients (r) are computed to justify the correlation between the dependent and independent variables.

The above two (Tables 1 and 2) present the correlation of all the values of (r) for the independent and dependent variables. The correlation matrix results have shown that most of the variables are positively correlated with one another in the pharmaceutical industry. The analysis, followed by the analysis, describes that the correlation matrix between the descriptive variables, i.e. ROA and ROI, is highly correlated, and the value is 0.984. Furthermore, the values of the correlation coefficients of ROE and ROI, ROE, and ROA are 0.932 and 0.919, respectively, and these variables are also highly correlated. Moreover, the investigation of the correlation matrix may indicate that the dependent variable EARPs is also significantly correlated with variables FFGC, SP, EF i.e., the values are 0.602, 0.560, 0.466, respectively.

On the other hand, in the textile industry, it is found that there is a solid significant and positive relationship between some of the variables, such as ROA and ROE is 0.548, ROA and ROI is 0.784, ROE and ROI is 0.823 respectively. Again, other variables also show a significant

Table 2. Correlation analysis of textile companies.

| Variables Name | FFGC | ROA | ROE | ROI | SP | EF | EARPs |
|---|---|---|---|---|---|---|---|
| FFGC | 1 | | | | | | |
| ROA | -0.007 | 1 | 1 | 1 | 0.796** | 1 | 1 |
| ROE | -0.066 | 0.548** | 0.823** | 0.112 | 0.615 | 0.709** | |
| ROI | 0.014 | 0.784** | -0.056 | -0.009 | | | |
| SP | 0.511** | 0.119 | -0.128 | -0.149 | | | |
| EF | 0.599** | 0.058 | -0.158 | | | | |
| EARPs | 0.315** | -0.142 | | | | | |

Source: SPSS output by analyzing secondary data.

Note 1: The dependent variable EARPs = Environmental Accounting and Reporting Practices. The independent variables are FFGC = Firms Features General Characteristics, ROA = Return on Assets, ROE = Returns on Equity, ROI = Return on Investment, SP = Stakeholder Pressure, EF = External Factors.

Note 2: *. Correlation is significant at the 0.05 level (2-tailed).

**. Correlation is significant at the 0.01 level (2-tailed).

positive relationship between them: FFGC and SP, FFGC and EF, SP and EF. Furthermore, the correlation between the dependent and independent variables shows a statistically significant relationship between EARPs and FFGC i.e. 0.315, and EARPs and EF i.e., 0.709. Nevertheless, in some cases, a non-significant negative relationship exists between the dependent and independent variables: EARPs and ROA is -0.142, EARPs and ROE is -0.158, and EARPs and ROI is -0.149.

The comparative synopsis of the correlation analysis of both the pharmaceutical and textile companies displays a diversified result. Pharmaceutical companies have revealed a significant positive relationship, whereas, in textile companies, the result is vice versa. Again, in the pharmaceutical companies' correlation matrix, there has no negative correlation between the variables. However, the scenario is different in textile companies, showing the statistically weak moderate and statistically non-significant negative relationship between the variables.

## 4.3 Regression analysis

To study any associative relationship and strength of associations between the selected independent variables i.e., FFGC, ROA, ROI, ROE, SP, EF with the dependent variable EARPs i.e., Index [33,113] regression analysis has been used. The results of the OLS model in (Tables 3 and 4) reveals that in the pharmaceutical industry, the intercept (constant) is -0.067and 0.135 in the textile industry, which infers that the expected value of dependent variable EARPs is average when independent variables in the model do not influence the dependent variable.

A regression coefficient of the independent variable indicates that for per unit (in percentage) increase in the independent variable, the average probable increase of dependent variable EARPs will be increased when other variables do not influence dependent variable EARPs.

So, the regression coefficient of independent variables FFGC i.e. 0.261, ROA i.e. 0.021 and SP i.e. 0.386 of the pharmaceutical companies indicates that for per unit (in percentage) increase in FFGC, ROA and SP the average probable increase of EARPs will be 0.261,0.021 and 0.386 when other variables do not influence EARPs. Again, the coefficient of EF changes per unit EARPs i.e. 0.051 and EF coefficient, is positive but has a statistically insignificant influence on EARPs. However, the independent variables' negative coefficient values, which signifies that per unit increase of negative coefficient of ROE -0.002 and ROI -0.019, decrease the EARPs when other variables have no influence on EARPs. The coefficients of ROE and ROI are negative as well as insignificant.

Table 3. Multiple regression model output of pharmaceutical companies.

| Variables Name | Unstandardized Coefficients | | Test Statistic (t) | Sig. Value (p) | Collinearity Statistics (VIF) |
|---|---|---|---|---|---|
| | B | Std. Error | | | |
| (Constant) | -0.067 | 0.067 | -0.992 | 0.325 | |
| FFGC | 0.261 | 0.094 | 2.786 | 0.007 | 2.281 |
| ROA | 0.021 | 0.009 | 2.260 | 0.028 | 3.792 |
| ROE | -0.002 | 0.003 | -0.899 | 0.372 | 4.883 |
| ROI | -0.019 | 0.010 | -1.884 | 0.065 | 7.206 |
| SP | 0.386 | 0.097 | 3.991 | 0.000 | 1.901 |
| EF | 0.051 | 0.101 | 0.507 | 0.614 | 2.071 |
| ANOVA | F Statistic: 11.167 | | Sig. Value (p): 0.000 | | |
| R | $R^2$ | Adjusted $R^2$ | Std. Error of the Estimate | Durbin-Watson Test Value | |
| 0.735[a] | 0.540 | 0.492 | 0.141 | 1.949 | |

Source: SPSS output by analyzing secondary data.

**Table 4. Multiple regression model output of textile companies.**

| Variables Name | Unstandardized Coefficients | | Test Statistic (t) | Sig. Value (p) | Collinearity Statistics (VIF) |
|---|---|---|---|---|---|
| | B | Std. Error | | | |
| (Constant) | 0.135 | 0.038 | 3.525 | 0.001 | |
| FFGC | 0.170 | 0.056 | 3.036 | 0.003 | 1.612 |
| ROA | -0.010 | 0.009 | -1.174 | 0.243 | 2.896 |
| ROE | 0.003 | 0.002 | 1.293 | 0.199 | 3.604 |
| ROI | -0.013 | 0.011 | -1.155 | 0.251 | 6.629 |
| SP | 0.157 | 0.084 | 1.875 | 0.064 | 2.952 |
| EF | 0.496 | 0.081 | 6.155 | 0.000 | 3.373 |
| ANOVA | F Statistic: 23.918 | | Sig. Value (p): 0.000 | | |
| R | $R^2$ | Adjusted $R^2$ | Std. Error of the Estimate | Durbin-Watson Test Value | |
| 0.769[a] | 0.592 | 0.567 | 0.130 | 1.931 | |

Source: SPSS output by analyzing secondary data.

Note 1: The dependent variable EARPs = Environmental Accounting Practices. Predictors: (Constant),

FFGC = Firms Features Genial Characteristics, ROA = Return on Assets, ROE = Returns on Equity, ROI = Return on Investment, SP = Stakeholder Pressure, EF = External Factors.

The fitted regression model based on statistical finding for the pharmaceuticals industry is as follows:

EARPs = (-) 0.067+0.261(FFGC) +0.021 (ROA) + (-) 0.002(ROE) + (-) 0.019(ROI) + 0.386 (SP) + 0.051(EF) + ε

Whereas, in the textile industry, a regression coefficient on FFGC is 0.170 for the textile companies, indicating that for per unit (in percentage) increase in FFGC, the average probable decrease of EARPs. Thus, the coefficient of FFGC is negative, but interestingly, it significantly impacts the EARPs. Additionally, the coefficient of EF changes the per unit EARPs in 0.496, and it has a statistically significant and robust influence on the EARPs. Following the analysis, the coefficient of SP is 0.157, ROE is 0.003, and signposts show that for per unit (in percentage), an increase in SP and ROE probably increases the EARPs. Further, SP and ROE show there is no impact on the EARPs. Afterward, the coefficient of ROA i.e. -0.010, and ROI, i.e. -0.013of the textile companies, indicates that for per unit (in percentage) increase in ROA and ROI, the average probable decrease of EARPs. The coefficients of ROA and ROI are negative and statistically insignificant, which have no influence on EARPs.

The fitted regression model based on statistical findings for textile industry as follows:

EARPs = 0.135+0.170(FFGC) +(-)0.010 (ROA)+0.003(ROE)+ (-)0.013(ROI)+ 0.157(SP) + 0.496(EF)+ ε

### 4.4 Result of the test of hypothesis

To test the significance of the relationship between the dependent variable (EARPs) and the independent variables (FFGC, ROA, ROE, ROI, SP, EF) described below (in Table 5):

## 5. Findings and discussion

After conducting the analyses outlined above, the results discussion will center on descriptive statistics, correlation, and Ordinary Least Squares (OLS) regression as the most reliable and efficient estimates based on the data of content analysis. The disclosure theory, emphasizing

Table 5. Result of alternative hypothesis.

| Hypothesis Description | Pharmaceutical Industry | | Textile Industry | |
|---|---|---|---|---|
| | P-value | Result for (Alternative Hypothesis) | P-value | Result for (Alternative Hypothesis) |
| H1: There is a relationship between Firm Features General Characteristics and EARPs | 0.007 | Supported | 0.003 | Supported |
| H2a: There is a relationship between ROA and EARPs | 0.028 | Supported | 0.243 | Not Supported |
| H2b: There is a relationship between ROE and EARPs | 0.372 | Not Supported | 0.199 | Not Supported |
| H2c: There is a relationship between ROI and EARPs. | 0.065 | Not Supported | 0.251 | Not Supported |
| H3: There is a relationship between Stakeholder Pressure and EARPs | 0.000 | Supported | 0.064 | Not Supported |
| H4: There is a relationship between External Factors and EARPs | 0.614 | Not Supported | 0.000 | Supported |

Source: Based on the analysis.

the divulgence of pertinent information in the realm of environmental accounting, encourages firms to enhance their environmental performance. Simultaneously, stakeholder theory anticipates that firms' reporting interests should mirror the needs of their stakeholders [114,115]. In testing the study's hypothesis, a significant relationship emerged between firm characteristics and Environmental Accounting and Reporting Practices (EARPs). The general characteristics of firms were positively and significantly linked to environmental accounting disclosure scores. The robustness of the results persisted through correlation analysis, with OLS regression yielding mixed outcomes in two industries. The regression equations shed light on the environmental practices in Bangladesh [94], aligning with stakeholder theory and confirming the positive relationship expected between firm characteristics and the quality of firms' environmental disclosure [10]. Nevertheless, limited empirical works have explored stakeholder theory's impact on reporting interests and documented the relative influence of stakeholders on the contents of firms' annual reports [114].

In the Appendix (Tables A1 and A2 in S2 File), descriptive statistics illustrate that eleven environmental indicators each exhibit a mean lower than 0.50 in both industries. However, the dependent variable EARPs indicated a mean value of 0.36 in the pharmaceutical industry and 0.30 in the textile industry, signifying poorer environmental procedures in the textile sector. Correlation analysis in the second phase revealed a positive and significant correlation coefficient between the independent variables of FFGC, SP, and EF with the dependent variable EARPs (Tables 1 and 2). Subsequently, regression analysis suggested that in the pharmaceutical industry, three independent variables FFGC, ROA, and SP significantly impact EARPs (Table 3). In the textile industry, two independent variables FFGC and EF demonstrated strong significance on EARPs (Table 4). These findings underscore a lower environmental practice in the textile industry compared to the pharmaceutical industry.

In the era of global warming and debates surrounding firms' environmental impact, accountability becomes crucial and can be promoted through integrated financial reporting. Given the substantial cost of reporting environmental activities, firms may not engage unless there are regulatory pressures or other formal mandates. However, in jurisdictions like Bangladesh, regulatory incentives for reporting environmental activities are still lacking, as reporting remains voluntary. Government intervention, as suggested by this study, could be a method to encourage such reporting. Contrary to the belief that better-performing firms contribute more to environmental practices, this study found no significant positive impact of organizational performance on EARPs. Instead, firm characteristics exhibited unique and significant impacts in both industries, aligning with previous literature [91]. Stakeholder and external pressures positively influenced EARPs in both industries, with SP having a significant impact in

pharmaceuticals and EP demonstrating a robust influence in textile companies, particularly in the export-oriented industry. Ultimately, the comparative results highlight that EARPs in Bangladesh are relatively lower than in developed nations, emphasizing the need for improvement for the long-term betterment of organizations.

## 6. Policy implications

The scope of accounting information users extends beyond shareholders and owners to encompass the general public. Consequently, the public holds the right to access information regarding companies' financial, environmental, and social initiatives, particularly when they impact their daily lives. This study on green accounting and reporting in Bangladesh's manufacturing industries, viewed through a holistic lens, holds significant implications for policy, industry practices, and corporate strategies. The findings underscore policymakers' need to refine and fortify environmental regulations, fostering collaboration with industry stakeholders to ensure effective implementation. Corporations are encouraged to embed environmental metrics into governance structures and management practices, aligning them with key performance indicators and executive compensation to incentivize sustainability. The report advocates for capacity-building initiatives for professionals, transparent stakeholder engagement, and the exploration of financial incentives to expedite the adoption of green practices. Again, this study sends messages to the government agency as well as the organization's owner, manager, activist stakeholder environmentalists, NGOs, and communities where these enterprises have been located, all of which may exert a positive attitude on the firms to practice EA and other environmental issues and focus more on external factors. Though, in the short run, firm performance does not influence the practice of EA in the long run, there would be a positive association between firm performances EAP as FFGC, external pressure, and stakeholder pressure positively impact EAP. The social implications of the outlined policy measures are profound, as they signify a potential shift in societal values and priorities. By promoting green accounting practices, the government is not only steering the business sector toward environmental responsibility but is also fostering a culture of sustainability within the broader society.

## 7. Conclusion, limitations, and future research guidelines

With the increase of globalization, the manufacturing sector of the globalized world increased its environmental and social responsibilities, [96,116]. The EARPs are concerned about the situation of Bangladesh, which is a vulnerable country. Based on content analysis, a rigorous search for indicators of green accounting directed from the GRI standards, and measuring the EARPs of the listed enterprises of Bangladesh, 175 observations examined from annual reports, the current study explored the factors affecting EARPs. According to a descriptive review of the firms' environmental features, both industries practice low EARPs. According to the findings, EARPs are substantially influenced by four variables: FFGC, ROA, SP, and EF. The study also shows that pharmaceutical companies perform better in EARPs than textile companies. Even though some factors are unrelated to the EARPs, businesses should implement environmental aspects as part of their self-interest. As a result, ongoing and strong actions are required to enforce environmental-related laws and regulations, particularly for industrial enterprises in these two sectors (pharmaceutical and textile).

The current analysis shows that the FFGC, FP, SP, and EP considerably directly impact the two most vulnerable sectors manufacturing firm's EARPs. This study supports the conclusion on existence of CG and CSR committee are positively associated with the firms EP and reflect sustainable development [117] also contradict with the findings of FFGC component CG mechanism affect firm performance but firm size are positively correlated with firm value.

Stakeholders' pressure has a substantial effect on Organizations motives and impacts on green management practice in emerging economies [118].

This research's most important theoretical contribution is that EARPs can vary by industry due to the various properties of FFGC, SP, and EP. This is a significant departure from prior material on the EARPs in emerging countries. This study suffers from several limitations that may influence its findings and must be considered in their analysis and interpretation. Due to the unavailability of data, this study had a small sample size, and discrepancies occurred, and several studies focused on the PLS-SEM method [119] to identify the main factors of social interaction and its growth.

In addition, this study elucidates various practical contributions for policymakers, such as the requirement for the firm to address environmental accounting practices difficulties in order to operate the business, which will support foreign investors to invest more in the emerging sectors. The GRI environmental standards would be firmly implemented through government activities, and businesses should follow the GRI recommendations for practicing environmental issues or developing their environmental standard in product manufacture.

Finally, while other aspects of EARPs cannot be explored in a single study, this work serves as a foundation for future research, and another key aspect of the study can be considered: how EARPs affect company performance. Although this study explores the pharmaceutical and textile industry scenario, the other manufacturing sectors can also be further focused on EAR practices.

## Supporting information

**S1 File.**
(XLSX)

**S2 File.**
(PDF)

## Author Contributions

**Conceptualization:** Mohammad Mizenur Rahaman.

**Data curation:** Mohammad Mizenur Rahaman, Adiba Rahman Bushra Chowdhury, Renhong Wu.

**Formal analysis:** Mohammad Mizenur Rahaman, Md. Alamgir Hossain.

**Investigation:** Mohammad Mizenur Rahaman, Renhong Wu.

**Methodology:** Mohammad Mizenur Rahaman, Shamima Akter, Md. Alamgir Hossain, Renhong Wu.

**Software:** Shamima Akter, Renhong Wu.

**Validation:** Shamima Akter, Adiba Rahman Bushra Chowdhury.

**Writing – original draft:** Mohammad Mizenur Rahaman, Md. Alamgir Hossain.

**Writing – review & editing:** Md. Alamgir Hossain, Adiba Rahman Bushra Chowdhury, Renhong Wu.

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
