## [Decision Letter · Decision Letter 0]

10 Jun 2024

PONE-D-24-16842Green Accounting and Reporting in Bangladesh's Manufacturing Industries: A Holistic PerspectivePLOS ONE

Dear Dr. Wu,

Thank you for submitting your manuscript to PLOS ONE. After careful consideration, we feel that it has merit but does not fully meet PLOS ONE’s publication criteria as it currently stands. Therefore, we invite you to submit a revised version of the manuscript that addresses the points raised during the review process.

We look forward to receiving your revised manuscript.

Kind regards,

Liu Yang

Academic Editor

PLOS ONE

Journal Requirements:

2. In the online submission form, you indicated that The data used to support the findings of this study are available from the corresponding author upon request.

3. We note that you have referenced Chang, H.C. which has currently not yet been accepted for publication. Please remove this from your References and amend this to state in the body of your manuscript: (Chang, H.C. [Unpublished]) as detailed online in our guide for authors

4. We notice that your supplementary tables are included in the manuscript file. Please remove them and upload them with the file type 'Supporting Information'. Please ensure that each Supporting Information file has a legend listed in the manuscript after the references list.

Reviewers' comments:

Reviewer's Responses to Questions

**Comments to the Author**

1. Is the manuscript technically sound, and do the data support the conclusions?

Reviewer #1: Yes

Reviewer #2: Yes

2. Has the statistical analysis been performed appropriately and rigorously? 

Reviewer #1: Yes

Reviewer #2: Yes

3. Have the authors made all data underlying the findings in their manuscript fully available?

Reviewer #1: Yes

Reviewer #2: Yes

4. Is the manuscript presented in an intelligible fashion and written in standard English?

Reviewer #1: Yes

Reviewer #2: Yes

5. Review Comments to the Author

**Reviewer #1:** This article analyzes a topical issue, specifically by examining the determinants for developing activities that enhance environmental sustainability. According to the authors, green accounting become essential for articulating organizational strategies and making investment decisions. Then the disclosure of environmental activities in annual reports and other firm’s reporting is getting fundamental on corporate reporting. For that purpose, authors used a content analysis of the annual reports based on the themes established in Global Reporting Initiative guide lines.

The structure of the article is correctly chosen and it exhibits a logical concern. The hypotheses are supported by the literature reviewed. However, I have a few recommendations:

i. In the title of this article, the authors refer to manufacturing industries but just two industries are analyzed. Then, the title should be adjusted.

ii. When first use an abbreviation define it, for example EA and EP.

iii. Authors do not specify which are the sustainable development goals (SGD) in introduction page 2. I suppose they refer to the United Nations SGD. It should be indicated in the text.

iv. The maximum and minimum values for EARP in pharmaceutic and textile industries in page 9 of text are not consitent with the values documented in tables A1 and A2 in appendix. It must be corrected. Therefore, authors must check all the statistics. In addition, it would be useful if authors compared with statistics of other studies to be clear the magnitude of these values.

v. The estimations do not use panel data, but the sample varies in cross and time series.

vi. Some typos should be removed. For example, it is missing a T in Table 1, and in page 14 a u in ultimately.

**Reviewer #2: **Review Report of the Article Titled: Green Accounting and Reporting in Bangladesh's Manufacturing Industries: A Holistic Perspective

The study investigated the factors affecting the green accounting and reporting practices in Bangladeshi textile and pharmaceutical companies. It is a quantitative article and the quantitative analysis portion is acceptable, the article has few lacking in terms of its presentation and rationale.

The rationale for selecting only the textile and pharmaceutical industries is not clear. I agree that both of these industries have contributed to environmental pollution, some other industries are doing the same. Most of the manufacturing companies (even some service-oriented companies) are polluting the environment. Why did the authors exclude those? The special focus on these two industries needs to be justified. The description here is not convincing enough.

One of the weakest parts of this article is the literature review section. The authors could not identify the research gap convincingly. In terms of social and environmental accounting, Bangladesh is a well-researched context. Too many articles exist on the social and environmental accounting practices of Bangladeshi companies. Many of these articles highlighted the determinants of social and environmental disclosure. The authors need to rationalize the research gap by highlighting the novelty of this article more persuasively. Many of the important articles on the Bangladeshi context did not get referred. I think the literature review is incomplete. There should be a separate section reviewing the prior literature on Bangladesh.

The authors should use a theoretical framework to construct the hypotheses. There should be a separate section on the theoretical framework. Though, in the abstract, the authors mentioned “it draws upon disclosure theory”, it is not enough. Many new theories explain social and environmental accounting practices. Please go through the recent literature on this topic.

Other than these, there are some typos.

If these issues are addressed, the article can be published.

6. PLOS authors have the option to publish the peer review history of their article (what does this mean?). If published, this will include your full peer review and any attached files.

Reviewer #1: No

Reviewer #2: No

---

## [Author Response · Author response to Decision Letter 0]

23 Aug 2024

Dear Sir, 

Thank you for inviting us to submit a revised draft of our manuscript entitled, " Green Accounting and Reporting in Bangladesh's Pharmaceutical and Textile Industries: A Holistic Perspective" to your Journal Plos One. We express our unconditional gratitude for you valuable and constructive review of our article. Your feedback will guide us to strengthen the research work and overall quality of the paper. With utmost respect, we have incorporated all the corrections as proposed by the reviewers: 

Incorporation of SUGGESTIONS and COMMENTS

1. Is the manuscript technically sound, and do the data support the conclusions?

• RESPONSE: I formatted the whole paper with the journal policies and in our perspective it is technically sound. We also added the sufficient data support in conclusion section. 

2. Have the authors made all data underlying the findings in their manuscript fully available?

• RESPONSE: We hope that the edited section clarifies the theme. The data set and analysis all are available.

3. In the online submission form, you indicated that the data used to support the findings of this study are available from the corresponding author upon request.

• RESPONSE: We attached the data set, also analysis section in the manuscript has been checked. 

4. We note that you have referenced Chang, H.C. which has currently not yet been accepted for publication. Please remove this from your References and amend this to state in the body of your manuscript: (Chang, H.C. [Unpublished]) as detailed online in our guide for authors

• RESPONSE: Thank you for this suggestion. The details are given on the (Chang, H.C. [Forthcoming], Environmental Management Accounting Within Universities: Current State and Future Potential, RMIT University, 2007), reference section. 

5. Any changes to the reference list should be mentioned in the rebuttal letter that accompanies your revised manuscript. If you need to cite a retracted article, indicate the article’s retracted status in the References list and also include a citation and full reference for the retraction notice.

RESPONSE: Added the following references to find out the similarities. 

6. In the title of this article, the authors refer to manufacturing industries but just two industries are analyzed. Then, the title should be adjusted. 

Response: Green Accounting and Reporting in Bangladesh's Pharmaceutical and Textile Industries: A Holistic Perspective.

ii. When first use an abbreviation define it, for example EA and EP.

Response: Added in page 5, 

iii. Authors do not specify which are the sustainable development goals (SGD) in introduction page 2. I suppose they refer to the United Nations SGD. It should be indicated in the text.

Response: United Nations (UN) sustainable development goals serve as guidelines and targets that every nation must follow to fulfill its developmental ambitions

iv. The maximum and minimum values for EARP in pharmaceutical and textile industries in page 9 of text are not consistent with the values documented in tables A1 and A2 in appendix. It must be corrected. Therefore, authors must check all the statistics. In addition, it would be useful if authors compared with statistics of other studies to be clear the magnitude of these values.

Response: It checked, there was typos, in table section. We checked and the data presentation was also corrected. 

The present study measures the descriptive statistics of dependent and independent variables of companies' pharmaceutical and textile industries presented in the Appendix (Table A1 and A2). Appendix A (Table-A1, Table-A2) explains the pharmaceutical and textile industry's descriptive statistics. This study found that all the pharmaceutical and textile industry indicators of the dependent variable of environmental aspects (Energy, Effluents and Waste, Products and Services, Materials, Supplier Environmental Assessment, Biodiversity, Environmental Compliance, Health and Safety, Water, Grievance Mechanism and EARP) showed a lower mean of less than 0.50. 

Furthermore, to show all the environmental aspects or practices in one variable, this study has accumulated the eleven variables to the EARPs. The mean value of this variable is 0.30, and the standard deviation is 0.197 in the textile industry. Following, the mean value of EARPs is 0.36, and the standard deviation, is 0.196 in the pharmaceutical industry. The range of the maximum and minimum values of both the textile and pharmaceutical industries Cumulative EARP are subsequently sated between 0.81 and 0.06 and 0.00 and 0.81, respectively.

Consequently, all the independent variables of the study showed mixed results. As per the analysis and by reviewing the annual reports, it is found that in the textile and pharmaceutical industry, the mean value of the FFGC is 0.66 and 0.82, and the standard deviation is 0.305 and 0.284. Therefore, all the indicators of firm performance variables, i.e., ROA, ROE, ROI, also showed mixed results. Finally, the variables have mean values for both textile and pharmaceuticals SP 0.38 and 0.52, EF 0.86, respectively. Similarly, in the textile industry, all the study's independent variables reveal mixed results. However, the comparative analysis of descriptive statistics shows that textile companies have poorer results than pharmaceutical companies in every aspect. Similar result found in that EP have significant impact (R2 = 0.61) on green accounting practices have a significant positive effect through economic and environmental practices and EF partially mediates the relationship between green accounting and environmental performance, (Rahman & Islam, 2023).

v. The estimations do not use panel data, but the sample varies in cross and time series.

Response: 

Study mainly used content analysis to quantify the environmental accounting and reporting practices in pharmaceuticals and textile industries for last five years and organizational performance quantitative data (time series) collected from annual reports.

vi. Some typos should be removed. For example, it is missing a T in Table 1, and in page 14 a u in ultimately.

Response: Yes more than a few typos have been corrected also. 

7. The rationale for selecting only the textile and pharmaceutical industries is not clear. I agree that both of these industries have contributed to environmental pollution, some other industries are doing the same. Most of the manufacturing companies (even some service-oriented companies) are polluting the environment. Why did the authors exclude those? The special focus on these two industries needs to be justified. The description here is not convincing enough.

Response: Textile industries contribute to 20% of the global water pollution, resulting in detrimental effects on ecosystems through water surface acidification and eutrophication (Granskog et al. 2020). Another example of how the trade-off between sustainability, efficiency, and effectiveness goals is a clear manifestation of the management complexity that impacts pharmaceutical firms and the health systems overall (Aquino et al., 2018). The requirement for the pharmaceutical sector to supply a greater choice of pharmaceutical products while pursuing sustainable development is being driven by two factors: an aging global population and enhanced healthcare systems (Schneider et al., 2010). The research will close this gap by examining how green accounting techniques, especially in the environmental and economic spheres, might enhance environmental performance and energy efficiency. According to (Arslan et al. 2021) and (Rahman & Islam, 2023), businesses should prioritize green accounting practices and coordinate their economic, environmental, and social policies with the Sustainable Development Goals (SDGs) of the United Nations and other environmental criteria.

8. One of the weakest parts of this article is the literature review section. The authors could not identify the research gap convincingly. In terms of social and environmental accounting, Bangladesh is a well-researched context. Too many articles exist on the social and environmental accounting practices of Bangladeshi companies. Many of these articles highlighted the determinants of social and environmental disclosure. The authors need to rationalize the research gap by highlighting the novelty of this article more persuasively. Many of the important articles on the Bangladeshi context did not get referred. I think the literature review is incomplete. There should be a separate section reviewing the prior literature on Bangladesh.

Response: We added a new section and added the literature on Bangladesh perspective. Also the gap and rationale added in the introduction section. 

The environment has suffered from careless industrial practices, consumer apathy to the environmental repercussions of rising consumerism, and economic inequality across societal segments in Bangladesh. A lot of emphasis has been paid to the economic aspect of sustainability, particularly in the past 20 years. This indicates a research void that the paper aims to bridge. 

Sustainable accounting, energy consumption, and environmental sustainability, especially as they relate to chemical and pharmaceutical businesses operating in developing nations such as Bangladesh, (Rahman and Islam, 2023). Pharmaceutical and chemical industries in Bangladesh use particular green accounting techniques, such as emission tracking, which involves the measurement and monitoring of greenhouse gas (GHG) emissions and other airborne pollutants by the companies, (Fernando and Hor, 2017). Another approach is waste management and minimization, in which businesses evaluate the production and disposal of hazardous waste and put strategies into place to lower trash generation, enhance disposal techniques, and recycle waste when it is practical (Dhar et al. 2022). It has been discovered that Green Human Resource Management has a positive impact on job performance and job satisfaction because it increases employee engagement and motivation to enhance organizational performance in Bangladeshi pharmaceutical companies. Additionally, GHRM advances the theories of action, job performance, and ability to inspire (Alam et al, 2023). Results showed that business efficiency and environmental compliance have a significant and positive impact on sustainability. They also showed that environmental costs mediated the effect of ecological accounting on sustainability in pharma enterprises in Ghana (Wiredu et al, 2023). Similarly, due to institutional pressure, waste management had a positive and significant impact on the environmental sustainability of small and medium-sized businesses (Latifah & Soewarno, 2023). Energy-efficient practices can be encouraged as a practical way for manufacturing enterprises to achieve sustainability, and energy-efficient practices can improve green tax policies (Uddin et al. 2023). Vertically, customers that care about the environment are pushing the textile sector to reveal the environmental impact of their products throughout their life cycles and are placing greater demands on sustainable products (Niinimaki et al., 2020; Biswas et al. 2024 and Mondal et al, 2024). This study offers insightful information about the complex relationships that exist between environmental accounting, financially successful production, and sustainable practices in the pharma textile sector and favorably affect financial performance without having a greater impact on sustainable production in textile sector. 

9. The authors should use a theoretical framework to construct the hypotheses. There should be a separate section on the theoretical framework. Though, in the abstract, the authors mentioned “it draws upon disclosure theory”, it is not enough. Many new theories explain social and environmental accounting practices. Please go through the recent literature on this topic.

Response: 

Many theories have been examined in EARP, Institutional theory defines that an organization's behavior and practices can be influenced by the external environment, which includes stakeholder norms, values, and expectations (Guerreiro et al. 2006). The application of institutional theory can offer a valuable perspective in comprehending the impact of institutional forces on the adoption of green accounting practices and energy efficiency measures in Bangladesh, particularly in the chemical and pharmaceutical sectors (Rahman & Islam, 2023).

In the disclosure literature, theories have been established to account for the rationale behind the choice to provide further information. The results obtained are inconsistent, and the empirical evidence does not consistently support disclosure theories (Urquiza et al. 2010). It has two main objective, firstly, to validate the factors used and how the factors influences results. Whereas, the voluntary disclosure theory's (VDT) assumptions, financial control variables are being included in an increasing number of environmental disclosure studies (Guidry & Patten, 2012). Alternatively, (Sheth et al. 2011) have distinguished between two distinct facets of the economic dimension of sustainability, namely external stakeholders and cost reduction. Several studies resulted, favorable correlation between CSR disclosure and environmental performance, bolstering the voluntary disclosure idea (Lu and Wang, 2021; Clarkson et al. 2008). Another fact is "transparency fallacy", when corporate transparency is forced upon it by external stakeholders; transparency should instead be linked to increased engagement with environmental issues and ethical business practices (Gold & Heikkurinen, 2018). In order to strategically reduce information asymmetries in their business behavior and dispel concerns about their societal legitimacy, firms use sustainability reporting as a crucial tool of corporate communication. It suggest that companies that perform better in terms of the environment have an incentive to release more information to external stakeholders. This is done to communicate the firm's other positive attributes and unobservable proactive strategy. In this study disclosure theory used, to identify the GRI parameters are fully supported and the EARP have positive impact on financial performance. 

Stakeholder pressure is a crucial component of green accounting. To succeed and maintain ecological sustainability, relationships between businesses and stakeholders are essential. According to (Owen et al. 1997), stakeholder theory implies that companies should take into account the expectations and interests of all parties involved, not just shareholders. Employees, clients, suppliers, investors, regulators, and the larger society in which the business operates are examples of stakeholders (Parmar et al. 2010). According to the theory, companies can generate value by attaining sustainable long-term success while striking a balance between the demands and expectations of many stakeholders (Freudenreich et al. 2020). The stakeholders of pharmaceutical and chemical industries in Bangladesh are diverse and include workers, clients, financiers, vendors, and legislators (Deb et al. 2024). Companies must strike a balance between these stakeholders' divergent expectations and interests if they are to flourish. Employees could be worried about having a safe and healthy workplace, for instance. Ecological disclosure is typically inadequate whenever participants are not involved (Nzama et al., 2023). Previous research has demonstrated that information discrepancies can be reduced when a business shares financial information and other details with its stakeholders (Fonseka et al., 2024). Furthermore, one of the main forces behind how businesses respond to ecological concerns is stakeholder pressure for greener operations and strategies in the companies where they operate (Dura and Suharsono, 2022). There is a dearth of research on the impact of stakeholder pressure on ecological sustainability, with the majority of studies being carried out in industrialized countries. Pressure from stakeholders has a significant impact on implementing green managem

---

## [Editor Report · Decision Letter 1]

28 Aug 2024

Green Accounting and Reporting in Bangladesh's Pharmaceutical and Textile Industries: A Holistic Perspective

PONE-D-24-16842R1

Dear Dr. Renhong Wu,

We’re pleased to inform you that your manuscript has been judged scientifically suitable for publication and will be formally accepted for publication once it meets all outstanding technical requirements.

Kind regards,

Liu Yang

Academic Editor

PLOS ONE
---

## [Editor Report · Acceptance letter]

2 Sep 2024

PONE-D-24-16842R1 

PLOS ONE

Dear Dr. Wu, 

I'm pleased to inform you that your manuscript has been deemed suitable for publication in PLOS ONE. Congratulations! Your manuscript is now being handed over to our production team.

Kind regards, 

on behalf of

Professor Liu Yang 

Academic Editor

PLOS ONE